# Carboxyl-Rich Carbon Dots as Highly Selective and Sensitive Fluorescent Sensor for Detection of Fe^3+^ in Water and Lactoferrin

**DOI:** 10.3390/polym13244317

**Published:** 2021-12-09

**Authors:** Xinxin Wang, Yanan Zhao, Ting Wang, Yan Liang, Xiangzhong Zhao, Ke Tang, Yutong Guan, Hua Wang

**Affiliations:** 1College of Food Science and Engineering, Qilu University of Technology (Shandong Academy of Sciences), Jinan 250353, China; 201695011018@stu.qlu.edu.cn (X.W.); 10431211055@stu.qlu.edu.cn (Y.Z.); 1350641616@163.com (X.Z.); qlut_guanyutong@163.com (Y.G.); 2College of Biotechnology, Qilu University of Technology (Shandong Academy of Sciences), Jinan 250353, China; tingwang_97@hotmail.com (T.W.); tangk2001@163.com (K.T.); 3School of Materials Science and Engineering, Shandong University, Jinan 250061, China

**Keywords:** carbon dots, DFT calculations, Fe^3+^, lactoferrin, fluorescent sensor

## Abstract

As lactoferrin (LF) plays an essential role in physiological processes, the detection of LF has attracted increasing attention in the field of disease diagnosis. However, most current methods require expensive equipment, laborious pretreatment, and long processing time. In this work, carboxyl-rich carbon dots (COOH-CDs) were facilely prepared through a one-step, low-cost hydrothermal process with tartaric acid as the precursor. The COOH-CDs had abundant carboxyl on the surface and showed strong blue emission. Moreover, COOH-CDs were used as a fluorescent sensor toward Fe^3+^ and showed high selectivity for Fe^3+^ with the limit of detection (LoD) of 3.18 nM. Density functional theory (DFT) calculations were performed to reveal the mechanism of excellent performance for Fe^3+^ detection. Meanwhile, COOH-CDs showed no obvious effect on lactobacillus plantarum growth, which means that COOH-CDs have good biocompatibility. Due to the nontoxicity and excellent detection performance for Fe^3+^, COOH-CDs were employed as a fluorescent sensor toward LF and showed satisfying performance with an LoD of 0.776 µg/mL, which was better than those of the other methods.

## 1. Introduction

Lactoferrin (LF), one number of transferrin families, is a 78 kDa iron-binding glycoprotein that was primarily identified from bovine milk in 1939 [1]. It has been regarded as the first defensive barrier for human beings due to its significant antimicrobial property, immune modulation, and tumor growth inhibition [2]. Therefore, LF is widely used in food and medical fields, such as infant formula, dairy products, and health products [3]. Recently, the requirement of LF sharply increased, as many reports reported that LF could help to some extent in COVID-19 prevention [4,5,6]. However, the structure of LF can be easily destroyed and lose function during processing [7]. Therefore, LF detection is highly desirable to ensure the quality of products before products enter the market. 

Various methods have been developed to detect LF, including high-performance liquid chromatography (HPLC), high-performance liquid chromatography-tandem mass spectrometry (HPLC-MS), electrochemiluminescence (ECL), and enzyme-linked immunosorbent assay (ELLSA) [8,9]. The benefits of these methods include their high sensitivity, easy operation, and wide detection range. However, they usually require expensive equipment, laborious pretreatment, and long processing time [10]. Up to date, developing an accurate, rapid, and convenient method to detect LF still remains a challenge. 

LF presents a unique structure in space with a polypeptide chain folded into two globular lobes to provide a binding site for Fe^3+^, and the binding reaction between LF and Fe^3+^ is reversible [11,12,13]. Kanwar et al., reported that the Fe^3+^ content in LF directly affected anti-tumor activity as well as apoptosis and cytotoxicity [14]. Meanwhile, Choi’s group found that Fe^3+^ binding in LF had a pivotal influence on LF physiological function, which can control the survival and death of the cell [15]. These reports indicated that the Fe^3+^ embedded into LF determined the functional property of LF, and the detection of Fe^3+^ in LF could be used to analyze the biological activity of LF [16]. 

Fluorescent sensors based on carbon dots (CDs) have attracted much attention due to their good biocompatibility, low toxicity, low cost, and excellent detection performance [17]. Guo et al., developed a simple method for the preparation of N, P-doped carbon dots with highly selectivity and sensitivity toward Fe^3+^ to utilize in photocatalytic degradation application [18]. Li et al., prepared CDs with high detection efficiency for Fe^3+^ to be applied in bioimaging [19]. In our previous work, a novel N-doped carbon dots for Fe^3+^ detection was demonstrated and utilized as an “on–off–on” fluorescent sensor for *Streptococcus thermophilus* stain-screening [20]. However, few works related to LF detection with a fluorescent sensor have been reported up to now. In this work, we demonstrated the synthesis of carboxylic carbon cots (COOH-CDs) via the hydrothermal method with tartaric acid as the precursor. The structure, morphology, and optical properties of COOH-CDs were further analyzed. Moreover, COOH-CDs were used as a fluorescent sensor toward Fe^3+^ and exhibited high sensitivity and selectivity. Density functional theory (DFT) calculations were performed to study the excellent performance for Fe^3+^ detection. To evaluate the biocompatibility of COOH-CDs, a toxicity experiment was also carried out. Furthermore, COOH-CDs were employed as a fluorescent sensor toward LF and showed satisfied performance.

## 2. Experimental

### 2.1. Materials

Tartaric acid and inorganic salts (including FeCl_3_·6H_2_O, ZnCl_2_, CoCl_2_·6H_2_O, CrCl_3_·6H_2_O, CuCl_2_·2H_2_O, PbCl_2_, FeCl_2_·4H_2_O, MnCl_2_·4H_2_O, and BaCl_2_·2H_2_O were obtained from Shanghai Aladdin Bio-Chem Technology Co., China. Lactoferrin (extract from milk) was purchased from Shanghai Macklin Biochemical Co., Ltd, China. Lactobacillus plantarum strain was obtained from the Culture Collection in our lab. MRS medium was prepared according to the published literature. 

### 2.2. Characterization

A Nicolet iS 10 FTIR spectrometer (Thermo Fisher Scientific, Shanghai, China) was utilized to measure FT-IR spectra with the range from 4000 to 400 cm^−1^. Ultraviolet (UV) absorption spectra were collected from a Shimadzu UV-2600 spectrometer, Japan. Photoluminescence (PL) spectra were obtained from a Hitachi F-4500 fluorescence spectrophotometer (Hitachi, Tokyo, Japan). A JEOL JEM2100 electron microscope (JEOL, Tokyo, Japan) was applied to obtain High Resolution Transmission Electron Microscope (HRTEM) images. ESCALABXi+ (Thermo Fisher Scientific, Shanghai, China) was used to collect X-ray photoelectron spectra. A OLYMPUS CX41TV biological microscope (OLYMPUS, Tokyo, Japan) was used to perform optical microscopic examination.

### 2.3. Synthesis of COOH-CDs

COOH-CDs were prepared by the hydrothermal method (Figure 1). Tartaric acid (0.1 g) was dissolved in 10 mL of ethanol and transferred into a Teflon reactor. The reactor was placed in an oven at 180 °C for 10 h. After cooling to room temperature, the crude solution was first filtered through a 0.22 µm membrane filter to remove large particles and then dialyzed with a dialysis bag (MWCO: 500 Da) for 48 h to obtain a transparent solution. COOH-CDs were obtained as a pale powder by freeze drying at −50 °C for 16 h.

### 2.4. Toxicity Experiment

To evaluate the biocompatibility of COOH-CDs, a toxicity experiment was carried out [21]. Considering its application in lactoferrin detection, lactobacillus plantarum was selected as the model with viable count survival rate as the toxicity index. The reference sample was prepared using the plate smearing method by inoculating 1% lactobacillus plantarum into the MRS solid medium and culturing at 37 °C for 24 h. Simultaneously, 0.1 g/mL COOH-CDs solution was used to partly replace water (70%, 80%, and 90%) or glucose (10%, 20%, and 30%) to prepare testing samples. The toxicity of COOH-CDs was analyzed by comparing the change of viable count with the reference sample. 

### 2.5. Fluorescence Selectivity and Sensitive of COOH-CDs towards Fe^3+^ and Lactoferrin

First, 10 μL solutions (10^−3^ mol/L) with different kinds of metal ions were utilized to analyze the selectivity of COOH-CDs (0.1 g/L) toward Fe^3+^. The changes of COOH-CDs’ fluorescence intensity as Fe^3+^ concentration increased were measured under 317 nm to study the fluorescence response behavior. The fluorescence response behavior of COOH-CDs toward Lactoferrin was investigated by the same method. 

### 2.6. Theoretical Calculations

To reveal the fluorescent selectivity of COOH-CDs toward metal ions, density functional theory (DFT) calculations were utilized to study the interactions and electronic transition between COOH-CDs and different kinds of metal ions (M^n+^ = Fe^3+^, Fe^2+^, Cr^3+^, Mg^2+^, Ca^2+^, Mn^2+^, Mg^2+^, Co^2+^, Cu^2+^, Pb^2^, and Zn^2+^). DFT-based Becke’s three-parameter hybrid exchange functional and Lee–Yang–Parr correlation functional (B3LYP) using 6-31+G* basis set with the pseudopotentials SDD (for main-group metals) and Lanl2TZ (for transition metals) were performed to analyze the geometry optimizations of the COOH-CDs/M^n+^ complexes and corresponding monomers [22]. To ensure all complexes set in minima potential energy surfaces, frequency calculations were processed at the same level of theory. GAUSSIAN (09 Revision 01, Gaussian Inc, Wallingford, CT, USA) and Gauss View (5.08, Gaussian Inc, Wallingford, CT, USA) were used to observe all the electronic structure calculations [20].

To calculate the binding energy (BE) of COOH-CDs/M^n+^ complexes, a supramolecular approach was used with the assisted of a counterpoise (CP) procedure suggested by Boys and Bernardi to correct the basis set superposition error (BSSE) [23].
(1)$$BE=E_{complex}-\sum_{i=1}^{n}E_{i}$$

As shown in Equation (1), *E_complex_* represents the total energy of the complex; *E_i_* is the energy of the monomer (i.e., COOH-CDs and M^n+^ ions). Grimme’s (DFT-D3) dispersion correction was applied to quantify the role of dispersion energy in COOH-CDs/M^n+^ complexes.

## 3. Results and Discussion

### 3.1. Characterizations of COOH-CDs

The functional groups on the surface of COOH-CDs were analyzed by FT-IR spectroscopy in Figure 1a. The peaks centered at 3415 and 3327 cm^−1^ corresponded to the stretching vibration of OH generated by O-H and CO-OH [24]. For the aromatic, the peaks centered at 3119 and 1450 cm^−1^ were corresponding to the stretching vibration of unsaturated H and skeletal vibration. The absorption peaks at 1747 and 1082 cm^−1^ were attributed to the stretching vibrations of C=O and C-O-C [25]. These results indicated that abundant -OH and -COOH existed on the COOH-CD’s surface. The optical properties of COOH-CDs, including UV-vis absorption, fluorescence excitation, and emission spectra, were measured and shown In Figure 1b. The absorption peak of COOH-CDs at 319 nm was aroused by n–π* transitions of C=O on the surface of COOH-CDs [26]. The emission spectrum showed a peak with the maximum wavelength at 435 nm (fluorescence quantum yield was 3.62%) under 317 nm. The morphology and size distribution of the prepared COOH-CDs were further analyzed by HRTEM and AFM, and these are displayed in Figure 1c,d, respectively. Furthermore, the uniform size distribution was also measured. According to the above results, COOH-CDs exhibited a spherical shape with the average diameters ranging from 1.5 to 5 nm and the height around 3.3 nm. The HRTEM of prepared COOH-CDs exhibited clearly crystalline lattice fringes (0.21 nm) aroused by the (100) in-plane lattice of graphene, indicating that the COOH-CDs had a graphitic carbon core [27]. 

X-ray photoelectron spectra were measured to further reveal the composition of COOH-CDs (Figure 2). The three peaks of C 1s at 288.8, 286.4, and 284.4 eV corresponded to -C=O, C-O-C, and C-C, respectively [28,29,30]. The three peaks in the O 1s spectrum can be attributed to C-O (533.08 eV), C-O-C/C-OH (532.37 eV), and C=O (531.6 eV) [31,32]. The XPS data were consistent with the FT-IR results, which further confirmed that abundant hydroxy and carboxylic groups existed on the surface of the prepared COOH-CDs. 

### 3.2. Effects of Excitation Wavelength and pH on Fluorescent Emission

As shown in Figure 3a, COOH-CDs showed a typical excitation-dependent feature. With the increasing of excitation wavelength from 280 to 380 nm, the PL peak gradually red-shifted to a longer wavelength, indicating that the PL emission can be easily tuned by adjusting the excitation wavelength. This excitation-dependent phenomenon can be attributed to the interplay of the core and functional groups on the surface of COOH-CDs [33]. It is well known that pH plays an important role in chemical environments of living things; thus, the PL emission of COOH-CDs under different kinds of pH was further investigated [34]. As shown in Figure 3b, the PL emission of COOH-CDs exhibited a pH-dependent feature and strong PL emissions in all pH environments. The emission peak gradually red-shifted with the pH ranging from 1 to 11, and the intensity firstly increased from 1 to 5 and then declined from 5 to 11. These results can be attributed to the protonation and deprotonation process of COOH on the CDs’ surface [35]. COOH turned to COO^-^ with increasing pH and led to the formation of a delocalized π bond and the increase in n electrons, indicating that COOH-CDs can be used in any environment with pH ranging from 1 to 11.

### 3.3. Toxicity Study of COOH-CDs

The nontoxicity property of carbon dots made it superior to other fluorescent sensors, especially in biological sample detection. COOH-CDs solution was partly substituted with water or glucose to analyze the influence of COOH-CDs on the growth of lactobacillus plantarum. As shown in Figure 4a,b, the addition of COOH-CDs had no obvious effect on lactobacillus plantarum growth. Intriguingly, the survival rate showed an increasing trend with the addition of COOH-CDs. In Figure 4c, the survival rate presents a steady upward trend when COOH-CDs partly replaced water or glucose. This result implied that COOH-CDs is a wonderful carbon source for bacterium and has great potential in bacterium culture. Meanwhile, these results further confirmed that COOH-CDs are nontoxic, which means they can be developed as a fluorescent sensor for bioactive substance detect in practical application.

### 3.4. Fluorescent Selectivity and Sensitivity toward Fe^3+^

CDs have been widely used as a fluorescent sensor due to their strong emission, high fluorescent sensitivity, and low cost. The rich COOH groups on the surface of COOH-CDs can easily coordinate with metal ions, leading to a significant variation of fluorescent emission [36]. Therefore, the fluorescent selectivity and sensitivity of COOH-CDs toward different kinds of metal ions were further investigated (Figure 5a). It can be clearly seen that the prepared COOH-CDs had high selectivity toward Fe^3+^ than other metal ions. Furthermore, the PL intensity remarkably decreased with the increase in Fe^3+^ addition, indicating that COOH-CDs had high fluorescent sensitivity toward Fe^3+^ and can be developed as an excellent fluorescent sensor for Fe^3+^ detection. To further evaluate the sensitivity of COOH-CDs toward Fe^3+^, Stern–Volmer plots with quenching efficiency as the *Y*-axis and Fe^3+^ concentration as the *X*-axis were analyzed. As shown in Figure 5c, the plot showed an upward trend with the increasing concentration of Fe^3+^ in the range of 0–6.62 μM. Intriguingly, the plot was an approximate straight line with a good linear relationship (R^2^ = 0.9923) under low concentration (Figure 5d), which means that the COOH-CDs had great potential in Fe^3+^ fluorescent sensing detection. The correlation can be formulated with the following equation [20].
(2)$$1-\frac{I}{I_{0}}=K_{sv}\left[M\right]$$
where *I* and *I*_0_ represent the fluorescent intensity of COOH-CDs with and without Fe^3+^ respectively, *K_sv_* represents the quenching constant that can be calculated from the slope of Stern–Volmer plot, and [*M*] is the concentration of Fe^3+^.

To further evaluate the fluorescent sensitivity of COOH-CDs toward Fe^3+^, the limit of detection (LoD) was calculated by Equation (3), where *S_d_* stands for the intensity standard deviation of the blank samples measured 10 times, and *K_sv_* is the quenching constant of Stern–Volmer plots [37,38]. The LoD of COOH-CDs toward Fe^3+^ is determined to be 3.18 nM, which is lower than many related reports (Table 1). It indicates that COOH-CDs is an excellent potential candidate for Fe^3+^ detection.
(3)$$LoD=\frac{3S_{d}}{K_{sv}}$$

### 3.5. Theoretical Analysis for the Selectivity of COOH-CDs toward Metal Ions

In this paper, COOH functionalized coronene was selected as the model to further study the interactions property between COOH-CDs and metal ions. It can be seen from the optimized geometries of COOH-CDs/M^n+^ complexes in Figure 6 that metal ions were absorbed on the surface of COOH-CDs to form stable complexes (except for Cr^3+^) with different bonded properties. Obviously, Fe^3+^ (2.05 Å), Fe^2+^ (2.03 Å), Cu^2+^ (2.04 Å), and Co^2+^ (2.02 Å) were strongly bonded on the surface of COOH-CDs, for the bond distances were shorter than those of other metal ions [49]. The results also reflected the binding energies (BEs) of the COOH-CDs/M^n+^ complexes listed in Table 2, and the order of the stability for the COOH-CDs/M^n+^ complexes is Fe^3+^ >> Cu^2+^ > Co^2+^ > Zn^2+^ > Fe^2+^ > Mn^2+^ > Mg^2+^ > Pb^2+^ > Ba^2+^. Obviously, Fe^3+^ had the strongest interaction property toward COOH-CDs as the BE is the highest, indicating that COOH-CDs preferred to adsorb Fe^3+^ rather than other metal ions.

Furthermore, frontier molecular orbital (FMO) of COOH-CDs/M^n+^ complexes were also investigated to understand the effect COOH-CDs and metal ions charge. The isosurfaces of the HOMOs and LUMOs orbitals (showed in Figure 7) and the calculated highest occupied molecular orbitals (HOMO) and lowest unoccupied molecular orbitals (LUMO) energy gaps (Egs, listed in Table 2) were further analyzed. The isosurfaces of the HOMOs and LUMOs orbitals were used to depict the orbital interactions between COOH-CDs and metal ions. It was found that all the HOMOs mainly localized in the carbon plane of COOH-CDs, while LUMOs apparently transferred to metal ions. This phenomenon is especially obvious in the COOH-CDs/Fe^3+^ complex, where more than half of the LUMO transferred to Fe^3+^ ions to generate a stable system with high hybridization. According to the literature, lower Egs reflected the hybridization degree of metal ions with functionalized CDs [49]. Comparing with other metal ions, Fe^3+^ with the lowest Egs (1.87 eV) presented perfect hybridization with COOH-CDs. Combining the geometries and FMO results, it can be concluded that COOH-CDs preferred to absorb Fe^3+^ on its surface to form a stable complex rather than other metal ions, which was in good agreement with the fluorescent selectivity and sensitivity toward Fe^3+^. 

### 3.6. Fluorescent Sensing Performance of LF

As a natural functional protein, the binding capacity to Fe^3+^ endows lactoferrin (LF) antibacterial ability, ferric balance adjusting, immunity enhancement, and other functions [50]. As the binding reaction between LF and Fe^3+^ is reversible, fluorescent compound with strong binding capacity to Fe^3+^ can capture Fe^3+^ from lactoferrin and lead to the change of PL emission. Therefore, the prepared COOH-CDs with highly fluorescent selectivity and sensitivity toward Fe^3+^ were selected as the fluorescent sensor for LF detection. PL emission changes of COOH-CDs as the lactoferrin concentration increased were measured. As shown in Figure 8a, the PL intensity of COOH-CDs decreased with the increasing of lactoferrin concentration. The Stern–Volmer plot presented a straight trend with a good linear relationship (R^2^ = 0.9815) when the concentration of LF increased from 0 to 6.62 μg/mL. According to the Stern–Volmer curve for COOH-CDs toward LF, the LoD value was calculated to be 0.776 µg/mL, which is lower than those of other related methods (Table 3). It indicates that the prepared COOH-CDs have potential as a fluorescent sensor for LF detection.

## 4. Conclusions

In summary, we developed an efficient method for Fe^3+^ and LF detection using COOH-CDs as the fluorescent sensor. COOH-CDs exhibited high selectivity and sensitivity toward Fe^3+^ with an LoD of 3.18 nM. The results of theoretical calculations indicated that the excellent performance can be attributed to the binding energy between COOH-CDs and Fe^3+^ (−593.82 kcal/mol) being much higher than other ions, leading to a sensitive fluorescence response. Meanwhile, COOH-CDs showed no obvious effect on lactobacillus plantarum growth, which means that COOH-CDs have good biocompatibility. Due to the nontoxicity and excellent detection performance for Fe^3+^, COOH-CDs were employed as a fluorescent sensor toward LF and showed high sensitivity with an LoD of 0.776 µg/mL. This work can provide a new avenue to design a fluorescent sensor with high performance for LF in practical applications.

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
