# Peer review of "Carboxyl-Rich Carbon Dots as Highly Selective and Sensitive Fluorescent Sensor for Detection of Fe^3+^ in Water and Lactoferrin"

_polymers, 2021, doi:10.3390/polym13244317_

Round 1

Reviewer 1 Report

The manuscript presents an efficient method for Fe3+ and Lactoferin detection using COOH-CDs as the fluorescent material. The paper is relevant for scientific community and has original aspects. The mechanism was proven by theoretical calculations.

References are relevant for the developed work and actual.

--line 63, page 2: correct cots.

--please correct in line 286 fluorescent sensor with fluorescent material.

Please clearly and in detail describe the interference study, and revise accordingly Figure 5 a, because is not easily understandable.

Reviewer 2 Report

The paper “Carboxyl-rich Carbon Dots as Highly Selective and Sensitive Fluorescent Sensor for Detection of Fe3+ in Water and Lactoferrin” by Xinxin Wang, Yanan Zhao, Ting Wang, Yan Liang, Xiangzhong Zhao, Ke Tang, Yutong Guan and Hua Wang describes innovative bioanalytics. The basic novelty are carbon dots which were synthesized from tartaric acid. They possess a shell of carboxyl groups which preferably form complexes with Fe(III)-ions. These interesting nano particles were characterized by different methods such as optical absorbance, fluorescence, TEM, AFM and XPS. Furthermore, the selectivity to Fe(III)-ions was confirmed by theoretical calculations. Thus, these nano particles are an excellent probe to detect lactoferrin via its iron content.
All experimental activities are thoroughly described and discussed. The paper can be published as it is.